# Mechanically robust supramolecular polymer co-assemblies

Julien Sautaux[1], Franziska Marx [1], Ilja Gunkel [1], Christoph Weder [1]✉ & Stephen Schrettl [1]✉

Supramolecular polymers are formed through non-covalent, directional interactions between monomeric building blocks. The assembly of these materials is reversible, which enables functions such as healing, repair, or recycling. However, supramolecular polymers generally fail to match the mechanical properties of conventional commodity plastics. Here we demonstrate how strong, stiff, tough, and healable materials can be accessed through the combination of two metallosupramolecular polymers with complementary mechanical properties that feature the same metal-ligand complex as binding motif. Co-assembly yields materials with micro-phase separated hard and soft domains and the mechanical properties can be tailored by simply varying the ratio of the two constituents. On account of toughening and physical cross-linking effects, this approach affords materials that display higher strength, toughness, or failure strain than either metallosupramolecular polymer alone. The possibility to combine supramolecular building blocks in any ratio further permits access to compositionally graded objects with a spatially modulated mechanical behavior.

---

[1] Adolphe Merkle Institute, University of Fribourg, Chemin des Verdiers 4, CH-1700 Fribourg, Switzerland. ✉email: christoph.weder@unifr.ch; stephen.schrettl@unifr.ch

The assembly of molecular or macromolecular building blocks through noncovalent interactions such as hydrogen-bonding, host–guest interactions, or metal–ligand complexes furnishes supramolecular polymers[1–4]. The association of the building blocks can in many cases be reversed through stimuli that interfere with the bonding. The resulting disassembly into monomeric or oligomeric species makes the materials easily processable[5–7], leads to stimuli-responsive behavior[8–11], and enables functions such as healing or recycling[12–18]. A widely exploited design approach for supramolecular polymers is to end-functionalize telechelic building blocks with hydrogen-bonding or metal–ligand binding motifs so that they can assemble through linear chain extension[5–7,13,19–23]. The mechanical properties of the resulting materials are governed by the nature of the telechelic core and also phase-separation effects, notably the assembly of the binding motifs into crystalline or glassy hard phases that act as physical cross-links[5,13,21–25]. Many supramolecular polymers are based on telechelics with a low glass-transition temperature ($T_g$) and these materials, therefore, exhibit low stiffness (Young's modulus <100 MPa) and strength (<15 MPa)[5–7,13,21–23]. If glassy or semicrystalline telechelics are employed, their respective characteristics dominate the mechanical properties and the noncovalent binding only provides for a marginal gain[26–28]. The supramolecular assembly of multifunctional low-molecular-weight building blocks equipped with hydrogen-bonding motifs was recently reported to afford glassy materials with a high density of noncovalent cross-links[14]. As a result, such polymers combine high stiffness and low melt viscosity, but they are also very brittle.

In conventional polymers, brittleness can be reduced by rubber toughening[29], and this strategy was recently explored in hydrogen-bonded supramolecular copolymers formed by combining two monomers that assembled into rigid and rubbery domains, respectively[30]. However, as in previous attempts to create supramolecular copolymers[31–33], macrophase separation proved to be an issue. We show here that this problem can be overcome in metallosupramolecular polymers (MSPs). Metal–ligand-binding motifs can display much higher association constants than hydrogen-bonding motifs[5,25,34,35], and we demonstrate that this permits combining two otherwise immiscible building blocks in any ratio. The approach gives access to healable materials whose properties reflect an otherwise inaccessible synergistic combination of the individual components and which permit creating of compositionally graded objects whose mechanical behavior can be spatially controlled.

## Results and discussion

### Metallosupramolecular copolymers and thermal properties.

The materials investigated here are based on monomers featuring the widely used 2,6-bis(1′-methylbenzimidazolyl)pyridine (Mebip) ligand[36–38], which reliably coordinates transition metal ions, including $Zn^{2+}$, with which it forms reversible bifold complexes[39]. To create a rigid MSP network, a new low-molecular-weight trifunctional building block (TAB) carrying this ligand was prepared and assembled with $Zn^{2+}$ ions (Fig. 1). As the toughening component, we employed a MSP based on a telechelic poly(ethylene-co-butylene) macromonomer carrying the same ligand (BKB)[13,23]. After spectrophotometric UV–vis titrations (Supplementary Fig. 1) confirmed the complex formation between TAB or BKB and zinc-(II)-bis-(tri-fluoro-methane-sulfonyl)imide ($Zn(NTf_2)_2$), films of the individual MSPs (TAB:Zn and BKB:Zn) and their copolymers (TAB/BKB:Zn) were prepared by combining stoichiometric amounts of the metal salt and the respective monomer(s) in $CHCl_3$/$CH_3CN$, solvent-casting and drying (see "Methods"). Interestingly, TAB/BKB mixtures with substoichiometric amounts or without the metal salt phase-separated (Supplementary Fig. 2).

Films with a thickness of 100–200 μm were produced by compression molding solvent-cast samples above 180 °C to ensure a consistent processing history before their thermal and thermomechanical properties were assessed by differential scanning calorimetry (DSC) and dynamic mechanical analyses (DMA). The DSC traces of TAB:Zn show a reversible melting transition ($T_m$) at ~220 °C and a glass-transition temperature ($T_g$) of ~140 °C (Fig. 2a and Supplementary Figs. 3 and 4), indicating the formation of a semicrystalline MSP. As previously reported[13,23], the DSC trace of BKB:Zn displays a $T_g$ associated with the poly(ethylene-co-butylene) core at around –50 °C and a melting transition associated with a crystalline phase formed by the metal–ligand complexes at ~260 °C (Fig. 2a and Supplementary Fig. 3). At first glance, the DSC traces of the TAB/BKB:Zn copolymers show a combination of the transitions observed for the two individual MSPs (Fig. 2a and Supplementary Fig. 3). Above 20 wt% of BKB, the glass transition of the poly(ethylene-co-butylene) phase is discernable and the signal grows with the BKB content. Similarly, the melting transition of crystalline TAB:Zn domains at ~220 °C is observable when the TAB fraction is 50 wt% or higher. Interestingly, however, instead of the weak melting transition associated with BKB:Zn (~260 °C), a more prominent peak at ~271 °C is observed for the copolymers, which indicates the formation of a mixed crystalline phase involving ligands of both monomers (Fig. 1). The crystallinity of neat TAB:Zn, BKB:Zn, and TAB/BKB:Zn with 50 wt% TAB depends on the processing history and is slightly lower in melt-processed than solution-cast samples (Supplementary Figs. 5–7). DSC cooling traces recorded immediately after heating samples to 300 °C display the expected crystallization events, and the second heating traces show that all transitions are reversible (Supplementary Fig. 3). However, prolonged heating to temperatures above 280 °C under ambient atmosphere causes degradation (Supplementary Figs. 8 and 9), although thermogravimetric analyses put the 5 wt% weight loss to above 350 °C (Supplementary Fig. 10).

### Thermomechanical properties of the copolymers.

The DMA trace of neat TAB:Zn is characteristic of a semicrystalline polymer with two slanted plateaus that are separated by a $T_g$ at ~140 °C (Fig. 2b, Supplementary Figs. 11–13, and Supplementary Table 1). The failure temperature of ~220 °C matches the $T_m$ established by DSC and the MSP's viscosity is reduced from 1.13 MPa s at 200 °C to 827 Pa s at 230 °C (Supplementary Fig. 14), corroborating a solid-to-fluid transition upon melting. At room temperature (25 °C), the material displays a storage modulus $E'$ of 1.4 ± 0.07 GPa, reflecting its rigid nature below the $T_g$. The DMA of BKB:Zn shows a glassy regime below $T_g$, which the maximum of the tan delta trace puts at –28 °C (Supplementary Fig. 11). The glassy regime is followed by a broad rubbery plateau with an $E'$ of 45 ± 2 MPa at 25 °C (Fig. 2b, Supplementary Figs. 11–13, and Supplementary Table 1). $E'$ decreases above ~140 °C, and mechanical failure occurs at ~260 °C, also around $T_m$. The DMA traces of the TAB/BKB:Zn copolymers reveal thermomechanical properties that reflect weighted contributions of the two components and, importantly, also a contribution from a mixed crystalline phase. Below the poly(ethylene-co-butylene) $T_g$, all samples are rigid and display an $E'$ of 1.8–2.4 GPa (Fig. 2b, Supplementary Figs. 11–13, and Supplementary Table 1). In the case of copolymers with ≥20 wt% BKB, the modulus drops above this temperature to an extent that reflects the content of BKB:Zn. Another step-wise modulus reduction is seen around the $T_g$ of TAB:Zn, which in turn is proportional to the content of the rigid MSP. All samples fail at their respective melting temperature (Supplementary Table 1). Importantly, materials whose DSC

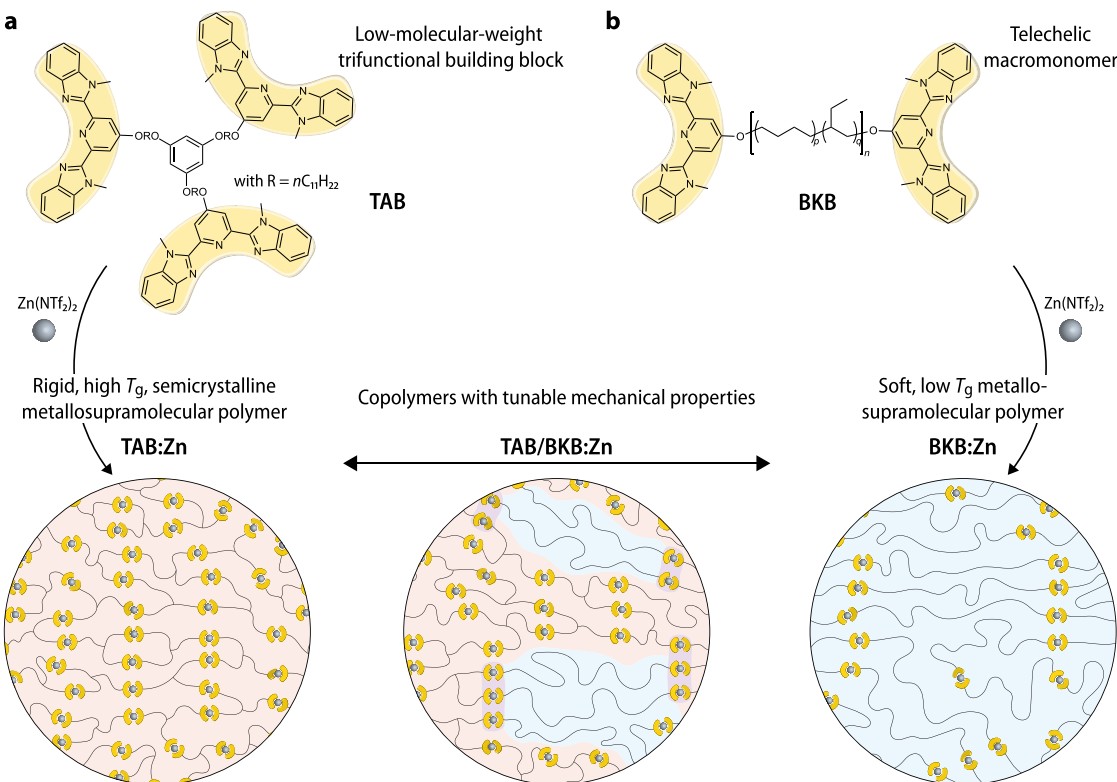

**Fig. 1 Structure of metallosupramolecular copolymers.** The individual metallosupramolecular polymers were assembled by combining $Zn(NTf_2)_2$ with (**a**) a trifunctional building block based on a 1,3,5-tris(alkyl)benzene core and three 2,6-bis(1′-methylbenzimidazolyl) pyridine (Mebip) ligands (TAB) or (**b**) a telechelic poly(ethylene-*co*-butylene) featuring two Mebip ligands (BKB). The co-assembly of different weight fractions of the two building blocks in the presence of stoichiometric amounts of zinc(II)bis(trifluoromethanesulfonyl)imide ($Zn(NTf_2)_2$) afforded metallosupramolecular copolymers TAB/BKB:Zn that feature a phase involving ligands of both monomers (purple) as well as domains of the individual TAB:Zn (orange) and BKB:Zn (blue) polymers.

traces display a signal associated with the crystalline phase involving both monomers mechanically fail at a higher temperature than the individual MSPs. Taken together, the DSC and DMA data suggest the formation of microphase-separated copolymers, whose thermomechanical properties are governed by well-defined TAB:Zn and BKB:Zn domains and a crystalline phase that involves both monomers.

**Copolymer microstructure.** Small- and wide-angle X-ray scattering (SAXS/WAXS) experiments were conducted to investigate the morphology of the various compositions (Fig. 2c and Supplementary Figs. 15 and 16). The scattering profiles of TAB:Zn corroborate the formation of a semicrystalline polymer network that adopts a lamellar morphology with a characteristic spacing of ca. 3.3 nm (Supplementary Fig. 17). As previously reported, BKB:Zn features a well-defined microphase-separated lamellar morphology with a characteristic spacing of ca. 9.0 nm, in which the metal–ligand complexes form hard domains that segregate from the poly(ethylene-*co*-butylene)[13,23,40]. The SAXS profiles of TAB/BKB:Zn samples with a BKB content of 80–90 wt% closely resemble the one of neat BKB:Zn, but with smaller characteristic domain spacings (Supplementary Table 2). This implies a decreased spacing of the poly(ethylene-*co*-butylene) phase and suggests the formation of a semicrystalline metal–ligand phase with increased lamellar width involving ligands of both monomers (Supplementary Fig. 18). The scattering profiles of copolymers with a TAB:Zn fraction of 30 wt% or more show the characteristic patterns of both MSPs and corroborate microphase separation into TAB-rich and BKB-rich domains (Fig. 2c and Supplementary Figs. 15 and 18). Compared to the neat MSPs,

decreased peak widths in the SAXS range (<4 nm$^{-1}$) indicate larger grain sizes for the TAB-rich and BKB-rich domains in the copolymers. Moreover, sharp WAXS peaks at larger $q$ values (> 6 nm$^{-1}$; Supplementary Fig. 16) differ from those of the neat MSPs and reflect a defined molecular spacing within a mixed metal–ligand phase at the interface of TAB-rich and BKB-rich domains (Supplementary Fig. 18).

To corroborate these findings, temperature-dependent SAXS/WAXS experiments were carried out. TAB:Zn retains the lamellar morphology and crystalline order up to the $T_m$ of 220 °C (Supplementary Fig. 19), as previously also reported for BKB:Zn[23]. While BKB:Zn samples degrade above $T_m$ (Supplementary Figs. 8 and 9), a peak broadening for TAB:Zn at 260 °C indicates simultaneous melting and disordering (Supplementary Figs. 19 and 20 and Supplementary Table 3). The morphology of TAB:Zn is restored upon cooling, but increased peak widths suggest a decrease in structural order. The SAXS data of the TAB/BKB:Zn (50:50 wt/wt%) copolymer reflect a minor thermal expansion below the $T_m$ of TAB:Zn (Fig. 2d, Supplementary Figs. 21–23, and Supplementary Table 4). Upon further heating, the reflections associated with the TAB-rich domains are significantly reduced and absent in samples that were briefly (1 min) heated to 280 °C and cooled to 250 °C. The profile recorded at this temperature exclusively displays SAXS peaks of the BKB-rich phase and WAXS peaks related to the mixed crystalline phase. Upon subsequent cooling to 180 °C, the peaks of the TAB:Zn phase and, hence, the morphology of the TAB/BKB:Zn (50:50 wt/wt%) copolymer were fully restored (Supplementary Fig. 23). While the positions of the first-order peaks $q^*$ (7.39–7.48 nm) and $q^{\#}$ (3.19–3.26 nm) barely change, an analysis

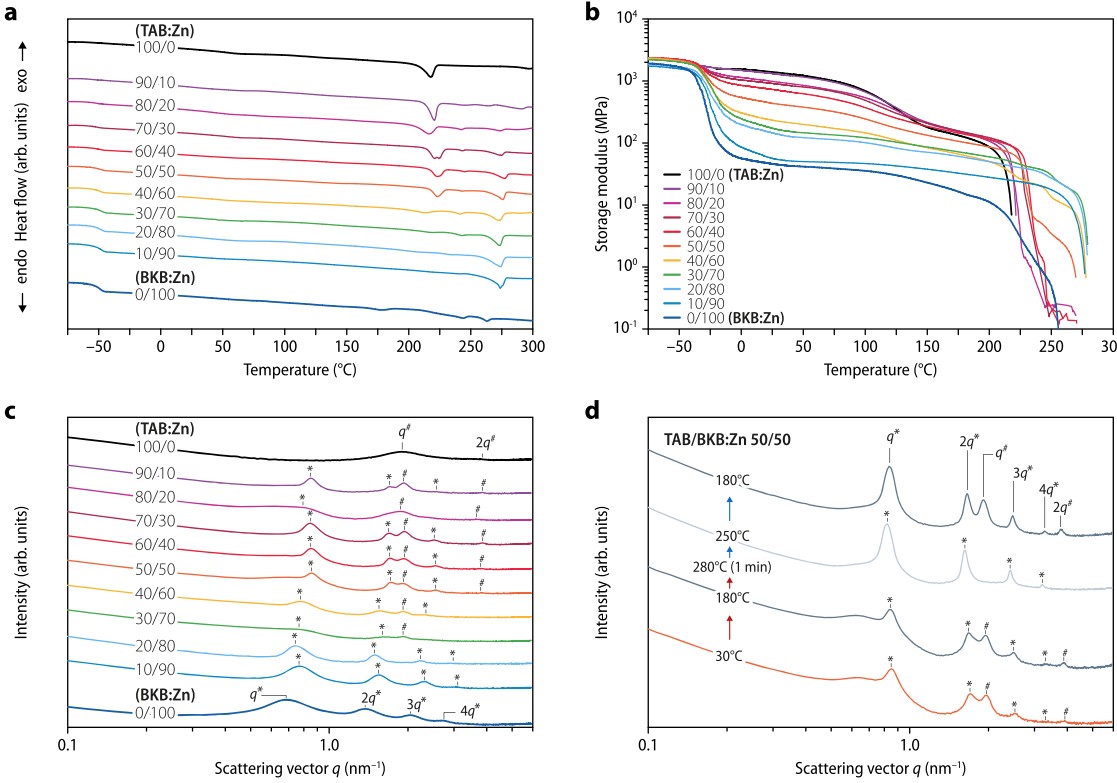

**Fig. 2 Thermal, thermomechanical, and structural properties of the MSPs and their copolymers. a** Differential scanning calorimetry (DSC) traces (first heating), **b** dynamic mechanical analysis (DMA) traces, and **c** small-angle X-ray scattering (SAXS) profiles of the neat metallosupramolecular polymers TAB:Zn and BKB:Zn, as well as of the TAB/BKB:Zn copolymers with the indicated TAB/BKB weight ratio (wt/wt%). **d** Temperature-dependent SAXS profiles of TAB/BKB:Zn containing TAB and BKB in a 50/50 wt/wt% ratio recorded at 30 and 180 °C, after cooling from the melt (1 min at 280 °C) to 250 °C, and after cooling to 180 °C. Heating and cooling rates were 10 °C min⁻¹. The DSC traces and scattering profiles are vertically shifted for clarity.

of their full width at half maximum suggests substantially increased grain sizes of both, BKB:Zn-rich (73–123 nm) and TAB:Zn-rich (32–48 nm) domains. Thus, the scattering experiments indicate that the copolymers can be thermally annealed to improve structural order, and that a mixed crystalline phase at the domain interfaces impedes macrophase separation.

The effect of annealing was further explored with the neat MSPs and select copolymer compositions. A comparison of the scattering profiles of solvent-cast, melt-processed, and quenched TAB:Zn samples shows that fast cooling decreases structural order and crystallinity (Supplementary Fig. 24 and Supplementary Tables 5 and 6). However, annealing of melt-processed samples at 180 °C, i.e., between $T_g$ and $T_m$, restores the order and increases the grain size (36–48 nm). The DMA traces show that the processing variations barely affect the moduli below ~100 °C (Supplementary Fig. 25 and Supplementary Table 7). Above $T_g$, quenched samples exhibit a lower $E'$ than slowly cooled materials, but the modulus increases again due to crystallization. Melt-processed TAB:Zn samples that were annealed (180 °C, 24 h) display a significantly increased modulus above $T_g$ (140 ± 3 to 427 ± 24 MPa at 170 °C) and a higher melting transition (212–224 °C). Annealing of compression-molded BKB:Zn (180 °C, 24 h) leads to an analogous increase of the grain size (37–93 nm) and crystallinity (Supplementary Figs. 26 and 27 and Supplementary Table 8). However, the DMA traces indicate that the improved order barely affects the mechanical properties. In line with these observations for the neat MSPs, scattering profiles of annealed (180 °C, 24 h) TAB/BKB:Zn samples with 20, 50, and 80 wt% of TAB indicate a general improvement of the structural order (Supplementary Fig. 26) and the DMA traces show an increase of the moduli above the $T_g$ of the TAB phase

(Supplementary Figs. 28–30). Optimized processing, hence, allows improving the structural order (Supplementary Figs. 31, 32 and Supplementary Table 9) and the thermomechanical properties at elevated temperatures by grain coarsening.

**Mechanical properties.** The mechanical properties of the two supramolecular polymers and their copolymers were further studied by uniaxial tensile tests at 25 °C, which were conducted with compression-molded samples and at a strain rate of 1% min⁻¹ (Fig. 3a, b and Table 1). The stress–strain curve of the neat TAB:Zn is characteristic of a rather brittle, rigid polymer, and reveals a Young's modulus of 1.11 ± 0.07 GPa, a tensile strength of 4.5 ± 0.3 MPa, brittle fracture at a strain of 0.41 ± 0.02%, and a toughness of 7.0 ± 0.9 kJ m⁻³. As previously reported, BKB:Zn exhibits plastic deformation, with a much higher elongation at break (116 ± 6%) and toughness (3580 ± 300 kJ m⁻³), but a lower Young's modulus (0.036 ± 0.001 GPa) than TAB:Zn[13]. The stress–strain curves of the TAB/BKB:Zn copolymers reveal that combining the two building blocks affords materials that cover a broad spectrum of mechanical properties. Importantly, it is possible to access materials that offer higher strength, toughness, or elongation at break than either of the neat MSPs (Fig. 3c and Supplementary Fig. 33). This emergent behavior contrasts the one of previously reported supramolecular copolymers[30,43]. On the BKB:Zn-rich end of the composition spectrum, the incorporation of a small amount (10 wt%) of TAB:Zn causes the extensibility and toughness to double, which is consistent with the observed increase of the crystalline hard phase, and thus the extent of physical cross-linking, vis-à-vis the neat BKB:Zn. Upon further increasing the TAB content to 40 wt%, the Young's modulus and

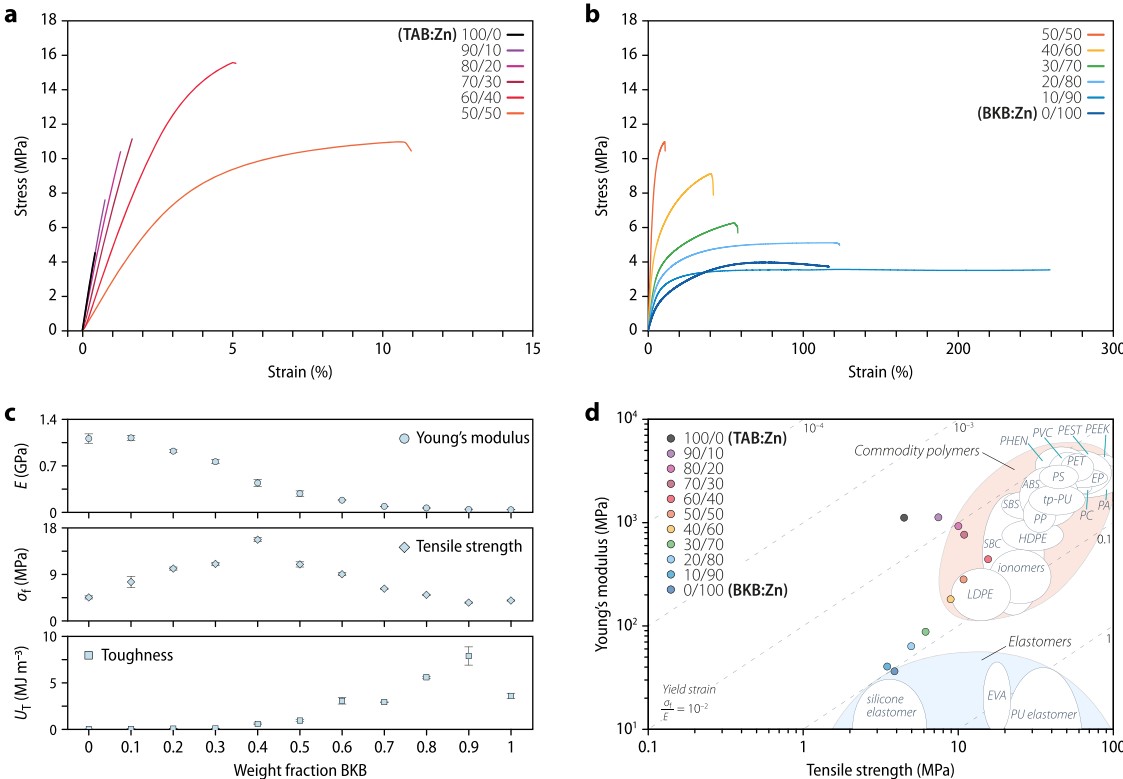

**Fig. 3 Mechanical properties of the two MSPs and their copolymers. a, b** Representative stress–strain curves of the metallosupramolecular polymers TAB:Zn, BKB:Zn, and their copolymers TAB/BKB:Zn with the indicated TAB/BKB weight ratio (wt/wt%). **c** Plots of the Young's modulus (E), tensile strength ($\sigma_f$), and toughness ($U_T$) of samples as a function of the weight fraction of BKB. Data represent averages of $n = 3–7$ individual measurements with standard deviation. **d** Ashby (materials selection) plot showing the Young's moduli and tensile strengths of the MSPs and copolymers and the properties attainable with some elastomers and commodity polymers[41, 42].

**Table 1 Overview of the mechanical properties.**

| Composition | Young's modulus (GPa)[a] | Tensile strength (MPa)[a] | Strain at break (%)[a] | Toughness (kJ m$^{-3}$)[a] |
|---|---|---|---|---|
| TAB:Zn | 1.11 ± 0.07 | 4.5 ± 0.3 | 0.41 ± 0.02 | 7.0 ± 0.9 |
| TAB/BKB:Zn (90/10) | 1.12 ± 0.04 | 7.5 ± 1.1 | 0.71 ± 0.13 | 27 ± 5 |
| TAB/BKB:Zn (80/20) | 0.92 ± 0.02 | 10.1 ± 0.3 | 1.2 ± 0.05 | 68 ± 4 |
| TAB/BKB:Zn (70/30) | 0.76 ± 0.03 | 11.0 ± 0.3 | 1.6 ± 0.03 | 92 ± 3 |
| TAB/BKB:Zn (60/40) | 0.44 ± 0.05 | 15.7 ± 0.4 | 5.6 ± 0.5 | 572 ± 93 |
| TAB/BKB:Zn (50/50) | 0.28 ± 0.04 | 10.9 ± 0.5 | 10.7 ± 1.1 | 935 ± 92 |
| TAB/BKB:Zn (40/60) | 0.18 ± 0.01 | 9.0 ± 0.3 | 42 ± 3 | 3060 ± 350 |
| TAB/BKB:Zn (30/70) | 0.087 ± 0.002 | 6.2 ± 0.1 | 58 ± 0.4 | 2919 ± 9 |
| TAB/BKB:Zn (20/80) | 0.063 ± 0.010 | 5.0 ± 0.1 | 125 ± 4 | 5585 ± 119 |
| TAB/BKB:Zn (10/90) | 0.040 ± 0.006 | 3.5 ± 0.1 | 238 ± 24 | 7887 ± 980 |
| BKB:Zn | 0.036 ± 0.001 | 3.9 ± 0.1 | 116 ± 6 | 3580 ± 300 |

[a]Data represent averages of $n = 3–7$ individual measurements ± standard deviation. Measured by stress–strain experiments at 25 °C with a strain rate of 1% min$^{-1}$.
Mechanical properties as determined by uniaxial tensile tests with the MSPs TAB:Zn and BKB:Zn and the copolymers TAB/BKB:Zn with the indicated TAB/BKB weight ratio (wt/wt%).

tensile strength are increased by factors of five and two, respectively, while a high toughness is maintained. This behavior reflects the presence of a reinforcing TAB:Zn hard phase as a minority component. On the other end of the composition spectrum, the TAB:Zn majority phase dominates the stiffness, while the incorporation of BKB:Zn has the intended toughening effect. Strikingly, the stress–strain curves not only show increased extensibility, but the tensile strength continuously increases up to a BKB:Zn content of 40 wt%, at which point it is tripled in comparison to TAB:Zn. Copolymers in the middle of the compositional spectrum display a tensile strength of 9–16 MPa and Young's moduli in the range of 0.2–0.4 GPa (Table 1). These

values are comparable to those of polyolefins[41,42], and put the metallosupramolecular copolymers in a property space that so far proved difficult to access with supramolecular polymers (Fig. 3d and Supplementary Fig. 34), although the deformation behavior differs.

**Healing metallosupramolecular copolymers.** The reversible assembly of BKB:Zn enables healing[13], and we demonstrate here that damaged samples of TAB:Zn and TAB/BKB:Zn with a 50:50 wt/wt% TAB:BKB ratio can also be healed. For light-induced healing, films of TAB:Zn were cut to a depth of ca. 30% of the

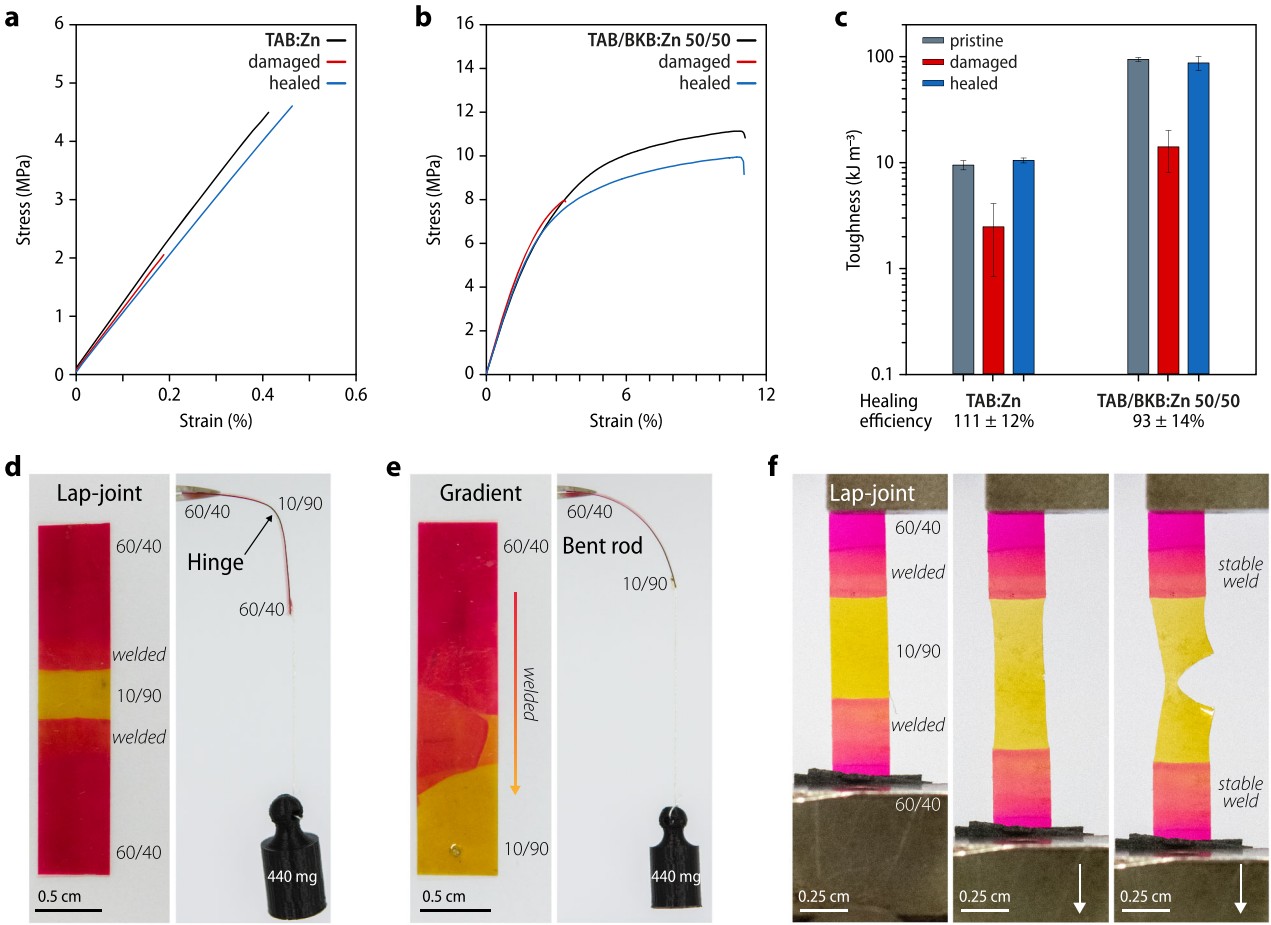

**Fig. 4 Healing and MSP copolymers with variable mechanical properties. a**, **b** Stress–strain curves of pristine, damaged, and healed (**a**) TAB:Zn and (**b**) copolymers with a TAB/BKB ratio of 50:50 wt/wt%. **c** Comparison of the toughness of pristine, damaged, and healed samples of TAB:Zn and copolymers with a TAB/BKB ratio of 50:50 wt/wt% (error bars represent the standard deviation of $n = 3–4$ individual measurements). **d** Photographs of a welded sample containing parts with TAB:BKB ratios of 60:40 wt/wt% (stiff, red-dyed) and 10:90 wt/wt% (soft, yellow-dyed). The soft segment acts as a hinge. **e** A compositionally graded film of MSP copolymers with TAB:BKB ratios of 60:40 wt/wt% (stiff, red-dyed) and 10:90 wt/wt% (soft, yellow-dyed) features a mechanical property gradient and bends under load. **f** The modulus of the welded sample shown in (**d**) is lowest in the segment with a TAB:BKB ratio of 10:90 wt/wt% (yellow-dyed) and the photographs show how the sample fails in this part upon uniaxial tensile deformation.

total thickness followed by exposing the damaged area to ultra-violet irradiation (320–390 nm; 320 mW cm$^{-2}$; see "Methods" for details). On account of light-to-heat conversion, a local temperature increase to ~215 °C was observed and optical micrographs show that the cut completely disappeared after 12 s (Supplementary Fig. 35). The efficacy of the process was quantified by uniaxial tensile deformation of pristine, damaged, and healed samples (Fig. 4a, c and Supplementary Table 10), and the data show that the mechanical properties of TAB:Zn were completely recovered. On account of the higher melting transition, healing of TAB/BKB:Zn films with a TAB:BKB ratio of 50:50 wt/wt% was slower, and heat-induced healing was explored to avoid long irradiation. Damaged samples were heated to 220 °C for different times and subjected to uniaxial tensile deformation (Supplementary Fig. 36; see "Methods" for details). A comparison of the stress–strain curves of pristine and damaged samples with those recorded after heating to 220 °C for 45 min shows that complete healing was achieved (Fig. 4b, c and Supplementary Table 11), demonstrating that the dynamic stimuli-responsive properties of MSPs are retained in the copolymers.

**Compositionally graded objects**. The possibility to combine two building blocks in any ratio and thereby form copolymers that cover a large property space is a feature that can be exploited to

fabricate objects in which the local mechanical characteristics vary in a precisely controllable manner. We demonstrate this by processing two different TAB/BKB:Zn samples with TAB:BKB ratios of 10:90 (dyed yellow) and 60:40 wt/wt% (dyed red) through compression molding above 180 °C into welded or compositionally graded films with stiffer and softer segments (Fig. 4, see Supplementary Method 1 for details). A simple segmented structure can serve as an all-polymer hinge (Fig. 4d), and deformation experiments show that the objects deform under load and failure eventually occurs where the strength is lowest (Fig. 4f and Supplementary Fig. 37). In a similar manner, a compositionally graded film was produced, which displays a mechanical property gradient that is reminiscent of (more complex) materials that connect hard and soft tissues in natural organisms (Fig. 4e)[44–48].

In summary, our systematic investigation of metallosupramolecular copolymers based on two strategically chosen building blocks has afforded materials whose mechanical properties can be controlled over a wide range, and which occupy a mechanical property space that so far remained difficult to access with supramolecular materials. The formation of microphase-separated morphologies featuring rigid and soft domains is the key aspect for the thermomechanical characteristics of these materials. Considering that macroscopic phase separation was

previously observed when combining hydrogen-bonded supramolecular polymers[30] suggests that binding motifs with high association constants similar to the one employed here are required to suppress this effect. The general concept of combining rigid and soft building blocks should be applicable to many supramolecular polymer systems, and emergent properties appear tangible.

## Methods

**Materials**. Spectroscopy grade $CHCl_3$ was purchased from Acros and purified from acidic impurities prior to use by passage through a plug of dry, activated (Brockman I) basic alumina. Zinc bis(trifluoromethylsulfonyl)imide (>97%, Strem Chemicals, Inc.), anhydrous $CH_3CN$ (Acros), spectroscopy grade $CH_3CN$ (Sigma Aldrich), and all other reagents and solvents (Sigma Aldrich) were used as received without further purification. The new low-molecular-weight trifunctional building block 1,3,5-tris((11-((2,6-bis(1-methyl-1$H$-benzo[$d$]imidazol-2-yl)pyridin-4-yl)oxy)undecyl)oxy)benzene (TAB) was prepared in three steps from commercial starting materials (Supplementary Fig. 38). A detailed description of the employed reagents and conditions and the analytical data for TAB and all intermediates are provided in Supplementary Methods 1–3. The 4-hydroxy-2,6-bis($N$-methylbenzimidazol-2'-yl)pyridine (Mebip) ligand and the telechelic bis(2,6-bis($N$-methylbenzimidazol-2'-yl)pyridine)-functionalized poly(ethylene-$co$-butylene) (BKB) macromonomer with a number-average molecular weight ($M_n$) of 4400 g mol$^{-1}$ were synthesized as previously reported[13,37].

**UV–vis spectroscopy and spectrophotometric titrations**. UV–vis absorption spectra were recorded on a Shimadzu UV-2401 PC spectrophotometer in $CHCl_3$/$CH_3CN$ (9/1 v/v). Titrations of TAB were carried out with $CH_3Cl$/$CH_3CN$ solutions ($c = 6.0$ μmol L$^{-1}$, 9/1 v/v, 2 mL) and addition of aliquots (25 μL) of a solution containing a mixture of $Zn(NTf_2)_2$ ($c = 126$ μmol L$^{-1}$) and TAB ($c = 6.0$ μmol L$^{-1}$) in $CH_3Cl$/$CH_3CN$ (9/1 v/v, 10 mL). Titrations of BKB were carried out with $CH_3Cl$/$CH_3CN$ solutions ($c = 9.4$ μmol L$^{-1}$, 9/1 v/v, 2 mL) and addition of aliquots (25 μL) of a solution containing a mixture of $Zn(NTf_2)_2$ ($c = 129$ μmol L$^{-1}$) and BKB ($c = 9.4$ μmol L$^{-1}$) in $CH_3Cl$/$CH_3CN$ (9/1 v/v, 10 mL). UV–vis spectra were recorded after the addition of each aliquot. To determine the stoichiometry for the metal–ligand complex formation, the absorption intensity at 340 nm of the metal–ligand charge transfer band was plotted against the metal-to-ligand ratio. UV–vis spectra of solid films of TAB and TAB:Zn with different amounts of the $Zn(NTf_2)_2$ metal salt were recorded in transmission on a Shimadzu UV-2401 PC spectrophotometer. Thin films (thickness <10 μm) were prepared by spin-coating (Spincoater Model P6700, Specialty Coating Systems, Inc.) ca. 5 μL of solutions of TAB (25 mg mL$^{-1}$) in $CHCl_3$/$CH_3CN$ (9/1 v/v) or TAB:Zn (25 mg mL$^{-1}$) in $CHCl_3$/$CH_3CN$ (9/1 v/v) with 0.5, 1.0, or 1.5 molar equivalents of $Zn(NTf_2)_2$ on quartz glass slides.

**Metallosupramolecular polymerization and film formation of TAB:Zn**. To a stirred solution of TAB (300 mg, 0.18 mmol) in $CHCl_3$ (10 mL), a solution of a stoichiometric amount of $Zn(NTf_2)_2$ (176 mg, 0.27 mmol) in anhydrous $CH_3CN$ (4 mL) was added dropwise and an increase of the viscosity was observed. The mixture was stirred for 30 min, cast into a poly(tetrafluoroethylene) (PTFE) Petri dish with a diameter of 6 cm, which was placed under a vacuum in an oven at 50 °C for 1 day. The sample was removed from the vacuum oven and a rigid, transparent film of TAB:Zn exhibiting appreciable strength and stiffness was obtained. Further compression molding of the material was carried out at either 220 °C or 180 °C in a Carver CE press to prepare films with a uniform thickness. For this, solvent-cast samples were placed between Kapton sheets that were kept apart by 100- or 200-μm-thick aluminum spacers. The temperature of the press was set to 220 °C and a pressure of 8 tons was applied to the samples for 10 s or a temperature of 180 °C was employed and a pressure of 8 tons was applied for 30 s. The metallic plates of the press were placed on the bench and the samples were left to slowly cool to room temperature between the plates. The typical cooling time after compression at 220 °C was ca. 45 min and ca. 30 min for cooling from 180 °C. This process furnished transparent films with a homogeneous thickness of either ca. 100 or ca. 200 μm.

**Metallosupramolecular polymerization and film formation of BKB:Zn**. The procedure for BKB:Zn mirrors the preparation of TAB:Zn. To a stirred solution of BKB (303 mg, 0.07 mmol) in $CHCl_3$ (5 mL), a solution of a stoichiometric amount of $Zn(NTf_2)_2$ (43.1 mg, 0.07 mmol) in anhydrous $CH_3CN$ (2 mL) was added dropwise. The mixture was stirred for 30 min, cast into a PTFE Petri dish (diameter 6 cm) and placed under vacuum in an oven at 50 °C for 1 day to obtain transparent films of BKB:Zn. Films of uniform thickness were prepared by compression molding between Kapton sheets as reported above at a temperature of 180 °C. A pressure of 8 tons was applied for 5 min, and the cooling time between the metallic plates was ca. 30 min. This process furnished homogeneous films of BKB:Zn with a thickness of either ca. 100 or ca. 200 μm.

**Metallosupramolecular polymerization and film formation of TAB/BKB:Zn copolymers**. The procedure for TAB/BKB:Zn copolymers mirrors the one for TAB:Zn. As a representative example, the preparation of TAB/BKB:Zn with a ratio of 50:50 wt/wt% is described in the following. To a stirred solution of TAB (103 mg, 0.06 mmol) and BKB (103 mg, 0.02 mmol) in $CHCl_3$ (15 mL), a solution of $Zn(NTf_2)_2$ (75.3 mg, 0.12 mmol) in anhydrous $CH_3CN$ (4 mL) was added dropwise. The mixture was stirred for 30 min, cast into a PTFE Petri dish (diameter 6 cm), and placed under a vacuum in an oven at 50 °C for 1 day. Films with a uniform thickness were prepared by compression molding in a Carver CE press. For TAB/BKB:Zn copolymers with ratios between 90:10–50:50 wt/wt% compression molding of samples was carried out between Kapton sheets at a temperature of 220 °C, whereas compression molding of samples of TAB/BKB:Zn copolymers with ratios between 40:60–10:90 wt/wt% was carried out between PTFE sheets at a temperature of 180 °C. In all cases, aluminum spacers with a thickness of 100 or 200 μm were used, a pressure of 8 tons was applied to the samples for 10 s, and the metallic plates of the press were placed on the bench to slowly cool samples to room temperature (ca. 45 min from 200 °C; ca. 30 min from 180 °C). Following this procedure furnished films of the TAB/BKB:Zn copolymers with a homogeneous thickness of either ca. 100 or ca. 200 μm.

**Thermal characterization**. Thermogravimetric analyses (TGA) were conducted under a nitrogen atmosphere in the temperature range of 25 to 600 °C with a heating rate of 10 °C min$^{-1}$ using a Mettler-Toledo STAR thermogravimetric analyzer. Differential scanning calorimetry (DSC) measurements were performed under a nitrogen atmosphere using a Mettler-Toledo STAR system operating at a heating/cooling rate of 10 °C min$^{-1}$ in the temperature range of −80 to 300 °C using a sample mass of ca. 5 mg, unless indicated otherwise. The midpoint of the step change in the heat capacity is reported as the glass-transition temperature $T_g$, and the melting temperature, $T_m$, is reported based on the minimum of the major endothermic melting peak.

**Thermomechanical and mechanical characterization**. All tests of the thermomechanical and mechanical properties were conducted on a TA Instruments DMA Q800 with rectangular-shaped samples with typical dimensions (length × width × thickness) of either ca. 20 × 5.3 × 0.2 mm or ca. 25 × 20 × 0.1 mm for dynamic mechanical analysis and tensile testing. Dynamic mechanical analysis (DMA) in the temperature range of −80 to 300 °C was conducted with samples fixed by tensile clamps under a nitrogen atmosphere with a heating rate of 3 °C min$^{-1}$, a frequency of 1 Hz, and an amplitude of 3 μm. Uniaxial tensile tests were also conducted at 25 °C with a strain rate of 1% min$^{-1}$. Young's moduli were determined from the entire curves, when samples displayed a linear relation between stress and strain until failure, otherwise they were determined from the slope in the linear region between 0 and 0.3% of strain. The reported mechanical data are averages of three to seven independent experiments, and all errors are standard deviations.

**X-ray scattering**. Small- and wide-angle X-ray scattering (SAXS/WAXS) experiments were performed with a NanoMax-IQ camera (Rigaku Innovative Technologies, Auburn Hills, MI, USA) equipped with a Cu target sealed tube source (MicroMax 003 microfocus, Rigaku). Scattering data were recorded by a Pilatus100 K detector (Dectris) and silver behenate was used as a reference to calibrate the sample-to-detector distance. To control the sample temperature, a Linkam HFS-X350-GI heating stage module with a T95 controller (Linkam Scientific) was used. For room-temperature scattering experiments, compression-molded polymer films were directly employed. For temperature-dependent scattering experiments, stainless-steel discs with a thickness of 2 mm and a central cylindrical hole (diameter of 2 mm) were employed as sample holders. To transfer samples into the sample holder, TAB:Zn was heated to 220 °C and transferred at this temperature. The heating and cooling rate for temperature-dependent scattering experiments was 10 °C min$^{-1}$, and samples were equilibrated at each temperature for 10 min before data collection was started. Scattering intensities are presented as a function of the momentum transfer $q = 4\pi\lambda^{-1} \sin(\theta/2)$, with the scattering angle $\theta$ and the photon wavelength $\lambda = 0.1524$ nm.

**Annealing experiments**. Annealing experiments with films of TAB:Zn, BKB:Zn, or TAB/BKB:Zn copolymers with different TAB:BKB ratios were carried out by placing compression-molded films with a homogenous thickness of either ca. 100 or ca. 200 μm in an oven at 180 °C for 24 h, unless otherwise noted. To thermally quench samples after compression molding, films were immediately removed from the metallic plates after compression molding at 220 °C and directly deposited on the bench to quickly cool to room temperature (ca. 10 min from 200 °C).

**Healing experiments**. Healing experiments were carried out with films of TAB:Zn with a uniform thickness of ca. 100 μm and films of TAB/BKB:Zn with a TAB:BKB ratio of 50:50 wt/wt% with a uniform thickness of ca. 200 μm, which were both prepared as described above by compression molding at 220 °C or 180 °C, respectively. Pristine samples were damaged by cutting to a depth of ca. 30 (TAB:Zn) or 40% (TAB/BKB:Zn 50:50 wt/wt%) of their total thickness with a razor blade attached to a caliper (to allow for depth control). For optical healing, samples

were placed on a substrate and irradiated with a Hoenle Bluepoint 4 Ecocure lamp connected to an optical fiber that was oriented normal to the sample surface at a distance of ca. 2 cm. An irradiation intensity of 320 mW cm⁻² was employed, and optical filters were used to either limit the output to the wavelength range of ultraviolet (UV) light between 320–390 nm or to the range of 390–500 nm. For thermal healing, samples were placed between two glass slides covered with Kapton sheets, clamped, and locally heated to 220 °C by placing the damaged sample region in a stream of hot air, while maintaining a distance of 8 cm between the sample and a hot air blower. The sample temperatures were monitored during optical and thermal healing experiments using an Optris PI connect infrared camera.

## Data availability

The datasets generated and analyzed during the current study are available from the corresponding authors upon request. The source data generated during this study and underlying Figs. 2–4, Table 1, Supplementary Figs. 1–38, and Supplementary Tables 1–11 are provided as a Source Data file and have been deposited in the Zenodo repository [https://doi.org/10.5281/zenodo.5720042]. Source data are provided with this paper.

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

## Acknowledgements

This material is based upon work supported by the Swiss National Science Foundation (Grant No. 200020_172619, C.W.) and the Adolphe Merkle Foundation. The authors

thank L.N. Neumann (Adolphe Merkle Institute), A. Petzold, and T. Thurn-Albrecht (Martin-Luther-Universität Halle-Wittenberg) for their help with viscosity measurements, S. Pal (Adolphe Merkle Institute) for help with the synthesis, and L.N. Neumann and B.D. Wilts (Adolphe Merkle Institute) for helpful discussions.

## Author contributions

J.S., C.W., and S.S. developed the original concept for the study and designed the materials and experiments. J.S. synthesized and characterized all materials and performed the experiments. F.M. synthesized and characterized samples with mechanical gradients, carried out healing and annealing experiments, and investigated melts. I.G. carried out and interpreted the X-ray experiments. All authors discussed the results and contributed to the interpretation of data. J.S., C.W., and S.S. wrote the paper. All authors contributed to editing the manuscript.

## Competing interests

The authors declare no competing interests.
