## [Peer Review File · Nature Communications]

Mechanically robust supramolecular polymer co-assembliesEditorial Note: This manuscript has been previously reviewed at another journal that is not operating a transparent peer review scheme. This document only contains reviewer comments and rebuttal letters for versions considered at *Nature Communications*. Mentions of the other journal have been redacted.

REVIEWERS' COMMENTS

Reviewer #2 (Remarks to the Author):

I have reviewed this manuscript earlier and was already (almost) satisfied with the previous version. In the current version the authors have answered my last question in an adequate manner and adjusted the text/figure accordingly. For me the difference in quality/impact between the journals [REDACTED] and [REDACTED] is very small, but since I was already satisfied with the last version, in my opinion this manuscript can be accepted as is.

Reviewer #3 (Remarks to the Author):

The authors provided an improved version of the manuscript and answered all my concerns. Since the manuscript is now transferred to Nat. Commun., I can support the publication of the results in the journal.

Revision of Manuscript NCOMMS-21-41006-T – Response to reviews

Referee #2

I have reviewed this manuscript earlier and was already (almost) satisfied with the previous version. In the current version the authors have answered my last question in an adequate manner and adjusted the text/figure accordingly. For me the difference in quality/impact between the journals [REDACTED] and [REDACTED] is very small, but since I was already satisfied with the last version, in my opinion this manuscript can be accepted as is.

We thank the reviewer for their very positive assessment of the revised manuscript and for supporting its publication.

Referee #3

The authors provided an improved version of the manuscript and answered all my concerns. Since the manuscript is now transferred to Nat. Commun., I can support the publication of the results in the journal.

We thank the reviewer for their valuable time and for supporting publication of the revised manuscript.